# Use of Gene Therapy in Retinal Ganglion Cell Neuroprotection: Current Concepts and Future Directions

**DOI:** 10.3390/biom11040581

**Published:** 2021-04-15

**Authors:** Jess Rhee, Kendrick Co Shih

**Affiliations:** 1Schulich School of Medicine & Dentistry, The University of Western Ontario, London, ON N6A3K7, Canada; jrhee2023@meds.uwo.ca; 2Department of Ophthalmology, Li Ka Shing Faculty of Medicine, The University of Hong Kong, Hong Kong, China

**Keywords:** retinal ganglion cell, optic nerve injury, gene-based therapy, glaucoma neuroprotection

## Abstract

We systematically reviewed published translational research on gene-based therapy for retinal ganglion cell (RGC) neuroprotection. A search was conducted on Entrez PubMed on 23 December 2020 using the keywords “gene therapy”, “retinal ganglion cell” and “neuroprotection”. The initial search yielded 82 relevant articles. After restricting publications to those with full text available and in the English language, and then curating for only original articles on gene-based therapy, the final yield was 18 relevant articles. From the 18 papers, 17 of the papers utilized an adeno-associated viral (AAV) vector for gene therapy encoding specific genes of interest. Specifically, six of the studies utilized an AAV vector encoding brain-derived neurotrophic factor (BDNF), two of the studies utilized an AAV vector encoding erythropoietin (EPO), the remaining 10 papers utilized AAV vectors encoding different genes and one microRNA study. Although the literature shows promising results in both in vivo and in vitro models, there is still a significant way to go before gene-based therapy for RGC neuroprotection can proceed to clinical trials. Namely, the models of injury in many of the studies were more acute in nature, unlike the more progressive and neurodegenerative pathophysiology of diseases, such as glaucoma. The regulation of gene expression is also highly unexplored despite the use of AAV vectors in the majority of the studies reviewed. It is also expected that with the successful launch of messenger ribonucleic acid (mRNA)-based vaccinations in 2020, we will see a shift towards this technology for gene-based therapy in glaucoma neuroprotection.

## 1. Introduction

Glaucoma is the leading cause of global irreversible blindness and is generally categorized into two broad categories: open-angle and angle-closure glaucoma [1,2,3]. Current treatment strategies for open-angle glaucoma include medical, laser, or incisional methods to lower intraocular pressure (IOP) [4]. These current methods to lower IOP are proven to slow the progression of glaucoma-mediated damage and are neuroprotective in nature. There has been slower progress in the development of IOP-independent neuroprotective and neuroregenerative strategies [5,6]. As current available treatments only address IOP, for ischemic and traumatic optic neuropathies, this treatment strategy is unable to address or reverse neuropathic damage [2]. Therefore, IOP-independent neuroprotective strategies would be beneficial for early stage intervention, potentially preventing disease progression, and reversal [4]. Retina ganglion cells (RGCs) are an important interventional target as glaucoma is characterized by optic nerve injury and the loss of RGCs [6]. Furthermore, aside from glaucoma, RGC death is central to other optic neuropathies, including demyelinating optic neuritis, ischemic optic neuropathy, and hereditary optic neuropathy [2]. The various etiologies of RGC death include defective axonal transport, ischemia, excitotoxicity, reactive oxygen species, trophic factor withdrawal, and loss of RGC electrical activity [4,7,8].

The underlying objectives of IOP-independent treatment strategies are to ameliorate optic neuropathies by providing an optimal environment for damaged RGCs and potentially regenerate damaged or dead RGCs into healthy functional ones [2,4]. These different strategies belong to three major categories, which include cell therapy, noncellular neuroprotective therapy, and gene delivery-based neuroprotective therapy. In this paper, the focus will be on gene delivery-based neuroprotective treatments, as gene-based therapeutic approaches have shown great promise in other central nervous system (CNS)-related injuries. The injection of an adeno-associated virus (AAV) that expressed NeuroD1 was able to convert reactive astrocytes in a severe stab injury model in mice into neurons, while the remaining astrocytes proliferated to repopulate themselves [9]. Similar findings were found in an in vivo model of adult non-human primates that utilized an ectopic injection of NeuroD1 AAV-based gene therapy in ischemic stroke monkey cortices [10]. Therefore, gene-based therapeutic approaches that could potentially reverse or provide neuroprotection against RGC death would address many optic neuropathies.

To determine the primary literature available focusing on gene therapy and RGC neuroprotection, the following search strategy was used in Entrez PubMed on 23 December 2020. The key terms used were “gene therapy”, “retinal ganglion cell”, and “neuroprotection”, with an “AND” between each keyword for a keyword search string of “gene therapy AND retinal ganglion cell AND neuroprotection”. Filters were used to limit the search to the last five years, full text publications, and English language papers only. This resulted in a total search hit of 82 articles. Figure 1 illustrates the screening process in a flowchart diagram. The articles were then curated to determine whether they were original research articles, which excluded 30 articles as they were review articles. The second curation was based on subject relevance—for example, a paper discussing outcomes after gene transfer using viral vectors for RGC damage would be included, whereas a paper that focused on the use of transgenic mice to study a mechanism of RGC injury would not be included. This second curation excluded 34 articles, leaving a total of 18 relevant articles for this review (see Figure 1).

## 2. Results

The search criteria and curating process used yielded 18 relevant papers that looked at the primary therapeutic and neuroprotective effect of gene therapy on retinal ganglion cells. From the 18 included papers, six of the studies utilized a viral vector encoding brain-derived neurotrophic factor (BDNF), two of the studies utilized a viral vector encoding erythropoietin (EPO), and the remaining 10 papers focused on different viral vector encoded genes and one microRNA study. The main results of each paper are summarized in Table 1, Table 2, Table 3, Table 4 and Table 5.

### 2.1. Adeno-Associated Viral (AAV) Vectors

The majority (17/18) of the papers utilized an adeno-associated viral vector as the platform for their gene therapy delivery. Table 1, Table 2, Table 3 and Table 4 summarize the one in vitro and 16 in vivo studies that were reviewed. The treatment outcomes and limitations of these papers will be summarized and categorized based on the gene delivery system used and the encoded gene.

### 2.2. AAV Vectors Encoding the BDNF Gene Therapy Approaches

BDNF is a neurotrophic factor that has functions both within and outside of the central nervous system and is able to regulate the survival, development, function, and plasticity of neurons [29]. BDNF is able to exert its functions by binding to its receptor, tropomyosin receptor kinase B (TrkB) [29]. One study utilized an adeno-associated viral vector (AAV2) encoding BDNF (AAV2-BDNF) that was injected unilaterally in their rat model [11]. Their model of injury induced elevated intraocular pressure (IOP) through microbead injections. Although microbead injections cause sustained IOP elevation and increase the cup/disc ratio, similar to glaucoma, the pathophysiology still remains different [30]. RGC neuroprotection was assessed through RGC count and there was a significant attenuation in RGC loss in injury + AAV2-BDNF treated rats compared to injury only rats. The unilateral treatment of BDNF injection led to a bilateral increase of BDNF in both eyes. There was a significant upregulation of BDNF protein levels in the injured + treated rats and a subsequent downregulation of TrkB protein levels. A limitation in this study was the injection of AAV2-BDNF three weeks prior to induction of IOP elevation, requiring specific timing to be utilized in a clinical setting. The stable expression of AAV vectors are established in two weeks, which highlights another temporal factor to consider for potential clinical applications [31]. A single intraocular injection of AAV-BDNF upregulated (approximately 2.5 fold) the expression of *bdnf* and lead to significant changes in gene expression in retinal genes, signal transduction, cell differentiation, and anatomical structure development [32]. As the expression of BDNF is not regulated in this system, the safety and implications of prolonged BDNF overexpression and its consequent changes in gene expression remain unexplored. Furthermore, overexpression of BDNF causes a counter downregulation of TrkB and a consequential desensitization of BDNF and its neuroprotective effects [33]. Therefore, regulation over the expression of these viral gene vectors is essential.

To address the downregulation of TrkB caused by the overexpression of BDNF, one group utilized a gene therapy consisting of an AAV2 vector encoding both BDNF and TrkB [12]. TrkB-2A-mBDNF was able to significantly attenuate immature SH-SY5Y human neuroblastoma cell loss and apoptosis when exposed to hydrogen peroxide. In mice with optic nerve crush, the same AAV2 TrkB-2A-mBDNF vector was able to significantly attenuate RGC loss more than BDNF treatment alone [13]. Treatment with both TrkB + BDNF had no change in glial fibrillary acidic protein (GFAP) levels, a marker of retinal stress, for up to 24 weeks, in *N*-methyl-D aspartate (NMDA) induced retinal injury. AAV2 TrkB-2A-mBDNF was able to significantly increase the positive scotopic threshold response (STR), but there was no significant difference in A- or B-wave responses. AV2-BDNF treatment was not considered in these set of experiments, however, BDNF treatment alone should have been an appropriate treatment group to demonstrate any protective visual function that BDNF treatment alone could have exerted and compared with TrkB + BDNF for any additive effect. Laser-induced ocular hypertension was used to transiently increase IOP in rats, and TrkB + BDNF was able to significantly attenuate axon loss and RGC loss. In both mice and rat injury models, the vectors were injected prior to injury and lacked any direct regulation in gene expression. The overexpression of BDNF and TrkB was maintained up to 24 weeks, highlighting the importance of long-term measurements on the safety and implications for possible clinical treatment.

Tyrosine triple mutant AAV (tm-AAV) vectors have increased efficacy for transduction into the retina [34]. One study utilized a self-complementary tm-AAV2 (tm-sc-AAV2) vector encoding BDNF (tm-sc-AAV2-BDNF) in an NMDA induced retinal injury mouse model [14]. This model of injury does not emulate the pathogenesis of glaucoma, as the retinal injury induced by NMDA is highly acute, while glaucoma is chronic. This study also pre-emptively injected their mice with tm-sc-AAV2-BDNF three weeks before the NMDA induced injury. The BDNF treated groups were significantly rescued from histological damage compared to injury only groups. Using an RGC marker, Brn3a, BDNF treated mice had a significantly greater number of Brn3a^+^ RGCs compared to NMDA treated mice. Visual function was assessed by scotopic ERG, and BDNF treatment had significantly higher B-wave amplitudes compared to the NMDA group. Treatment with BDNF was able to significantly reduce the levels of GFAP compared to injury only mice. This same tm-sc-AAV2-BDNF vector was utilized by another group on an ischemia/reperfusion injury (I/R) model, causing transiently increased IOP in rats [15]. The I/R injury was induced through the cannulation of normal saline until the IOP was raised to 110 mmHg for 60 min [35]. In this model of injury, the pathogenesis of glaucoma is not accurately portrayed as it is very transient and acute. The mice were injected with tm-sc-AAV2-BDNF two weeks prior to the induced retinal I/R injury. BDNF was able to significantly attenuate retinal thickness and structure loss, rescue RGCs, increase B-waves, and reduce GFAP expression. In both studies, the same tm-sc-AAV2-BDNF vector was used and although this vector has been shown to have greater transduction efficacy and accelerated protein expression, there was no regulation over the expression of BDNF with this vector [34,36].

Collapsin response mediator protein 2 (CRMP2) signaling is linked with the inhibitory effects of myelin on neurite outgrowth, and antagonizing CRMP2 signaling can promote axonal regrowth [37,38]. Mutation of the threonine 555 site to alanine (T55A) leaves CRMP2 phosphorylation resistant, thus limiting axonal degeneration and demyelination in mice [39,40]. One group utilized an AAV2 vector encoding this T555A mutation of the CRMP2 gene (AAV2-CRMP2T555A) and an AAV2 vector encoding both BDNF and CRMP2T555A (AAV2-BDNF + AAV2-CRMP2T555A) on rats with unilateral transection of the optic nerve (ON) [16]. A sham-operated group was not utilized as they stated that previous findings determined that there were no significant differences between sham-operated and normal rats in relevant outcomes and RGC numbers [16,41]. However, a sham-operated group would have been an appropriate control group to include. Rats were pre-emptively injected with treatment vectors 10 days prior to injury and outcomes were assessed three months later. Visual function as assessed by optokinetic nystagmus responses and total pursuits were significantly increased in BDNF, CRMP2T555A, and BDNF + CRMP2T555A treated groups. RGC counts were considered in dorsal, central, and ventral regions of the retina. There was no significant difference in RGC counts in the dorsal retina. In the central retina, BDNF, CRMP2T555A, and BDNF + CRMP2T555A treatment significantly increased βIII-tubulin^+^ RGCs, while only CRMP2T555A and BDNF + CRMP2T555A treatment significantly increased Brn3a^+^ RGCs. In the ventral retina, only AAV2-BDNF treatment significantly increased both Brn3a^+^ and βIII-tubulin^+^ RGCs. Axonal degeneration was significantly attenuated in BDNF, CRMP2T555A, and BDNF + CRMP2T555A treatment groups. Structural disruptions to the nodes of Ranvier were assessed by examining paranodal gaps, and BDNF and BDNF + CRMP2T555A significantly decreased the paranodal gaps induced by ON injury. The G-ratio is the ratio of inner axonal diameter to the total outer diameter used to assess axonal myelination [42]. The G-ratio increases following ON injury, and the BDNF, CRMP2T555A, and BDNF + CRMP2T555A treatment groups all significantly reduced the G-ratio. Only BDNF + CRMP2T555A was able to significantly reduce the oxidative stress marker, 4-hydroxynonenal (HNE) [43].

In different in vivo and in vitro animal studies as well as different eye injury models, AAV vectors encoding BDNF have demonstrated consistent and considerable neuroprotection of RGCs and visual function preservation [11,12,13,14,15,16]. Direct intraocular injections of BDNF have already been established to significantly increase the survival of RGCs, and so the demonstration that these gene-based therapies show similar results is a promising next step to potential clinical usage [29].

### 2.3. AAV Vectors Encoding the EPO Gene Therapy Approaches

EPO has been shown to exert neuroprotection in models of neuron damage and loss [44,45]. In a clinical trial, EPO was able to significantly reduce infarct size and improve functional outcomes in acute ischemic stroke patients [46]. One group utilized a *N*-methyl-*N*-nitrosourea (NMU) induced model of retinal degeneration with rapid progression in a mouse model that was treated with an AAV2 vector encoding EPO (AAV-2/2-mCMV-EPO) [17]. Mice were injected with AAV-2/2-mCMV-EPO three weeks prior to MNU induced injury and assessed one week later. Treatment with EPO resulted in significantly thicker retinas compared to injured mice. The number of cone photoreceptors was also significantly rescued in EPO treated mice compared to injured mice. Visual function was assessed with ERG and scotopic B-wave amplitudes, and ERG responses were significantly higher in EPO treated mice compared to injured mice. To determine if this protection in visual function offered any behavioral benefit, vision-guided optokinetic tests were assessed. Visual acuity and contrast sensitivity were significantly higher in the EPO treated mice compared to injured mice. Multi-electrode arrays (MEA) were used to detect the firing activities of RGCs, which were categorized into central, mid-peripheral, and peripheral distances from the optic nerve head. EPO treatment was able to significantly increase the amplitude of field potential compared to injured mice, but the results were not homogenous. EPO treatment had the greatest significant effect in the peripheral region, and the mid-peripheral region was significantly larger than the central region. An increase in the spontaneous firing rate of RGCs indicates hyperactivity that occurs during retinal degeneration. EPO treatment was able to significantly attenuate the spontaneous firing rates. The proposed mechanism for the neuroprotective effect of EPO was suggested to be partially anti-apoptosis mediated as the mRNA levels of apoptotic factors, caspase-3, CHOP, and Bax, were significantly downregulated in EPO treated mice, whereas anti-apoptotic factor, Bcl-2, was significantly upregulated. The NMU induced model of retinal degeneration has a different pathophysiology than that of clinical retinitis pigmentosa, which is chronic. Furthermore, the retinal degeneration that occurs in MNU-induced retinal degeneration leading to photoreceptor cell death is different from those in genetic or light-damaged retinal degeneration models [47,48]. Therefore, the subretinal delivery of EPO may not be effective for all retinal degeneration [17]. The administration of systemic EPO increases hematocrit and the risk of thrombosis, which is important to consider [49]. The eye is considered a closed system and intraocular EPO injections are assumed to carry less risk than systemic administration, although the long term risks and effects have not been studied.

Another group utilized a recombinant adeno-associated virus (rAAV) encoding EPO.R76E, a form of EPO with attenuated erythropoietic activity that has been shown to preserve RGC axons, cell bodies, and vision in the DBA/2J model of glaucoma [18,50]. The DBA/2J mouse progressively develops a mouse model of glaucoma, and these mice were injected at 5 months of age with EPO.R76E, intramuscularly. There was a significant increase in hematocrit in EPO.R76E treated mice at 4 weeks and 12 weeks post injection compared to GFP injected mice. The flash visual evoked potentials (FVEP) P1 peak and N1 peak amplitudes in EPO.R76E treated mice were not significantly different compared to negative or GFP controls, indicating a protection against vision loss. However, IOP continued to rise as these mice aged, even with EPO.R76E treatment. Microglia numbers and reactivity increased prior to the onset of elevated IOP in DBA/2J mice, and this alteration in microglia first occurred in the central and then the peripheral retina [51]. The number of microglia were significantly reduced in EPO.R76E treated mice compared to the GFP control in both the central and peripheral retina. The morphology of the microglia was also different in EPO.R76E treated mice, and the average soma area was significantly smaller compared to the GFP control, but significantly larger than the negative control in the central retina. There was no significant difference in the average soma area of microglia between EPO.R76E treated and GFP control mice in the peripheral retina. Further assessment of pro-inflammatory cytokines and chemokines showed that EPO.R76E treated mice had significantly lower mRNA levels of IL-1α, IL-1β, IL-17, IL-12p40, IL-12p70, CCL4 (MIP-1β), CCL5 (RANTES), and IL-13. There was also a trend of increased mRNA levels of antioxidant enzymes in EPO.R76E treated mice, but the trend was not significant. The vector used in this study reached stable gene expression in three weeks, again highlighting the temporal factor to consider in a clinical approach. The authors also stated that the effectiveness of this treatment was due to the long-lasting expression and EPO.R76E being safer than wild-type EPO, as it does not cause a dangerous rise in hematocrit [18]. However, as hematocrit levels were shown to be significantly elevated at weeks 4 and 12, and long-term effects were not assessed, the safety of systemic EPO.R76E treatment is still in question.

### 2.4. AAV Vectors Encoding the NRF2 Gene Therapy Approaches

Both nuclear factor (erythroid-derived 2)-like 2 (NRF2) and sirtuin 1 (SIRT1) can protect and prevent RGC loss in experimental autoimmune encephalomyelitis (EAE) induced optic neuritis [52,53]. NRF2 is a basic leucine zipper transcription factor that regulates the expression of antioxidant and detoxification enzymes [52]. SIRT1 is an NAD-dependent deacetylase that attenuates oxidative stress in neurons [53]. One group utilized AAV2 vectors encoding SIRT1 and NRF2 (AAV2-SIRT1 and AAV2-NRF2), separately, on EAE induced mice [19]. Mice were given intravitreal injections of AAV2-SIRT1 or AAV2-NRF2 four weeks prior to EAE induction with myelin oligodendrocyte glycoprotein peptide injection. The optokinetic response (OKR) was used to measure visual acuity and function in mice, and only AAV2-SIRT1 treatment was shown to significantly improve OKR compared to injury + GFP mice. Only AAV2-NRF2 treatment was able to significantly increase the number of Brn3a^+^ RGCs compared to injury + vehicle mice. However, this significant increase with NRF2 treatment was not significantly different from injury + GFP mice, which was not expected to have any effect. Both SIRT1 and NRF2 therapy were unable to significantly attenuate immune cell infiltration or ON demyelination. All groups with EAE induced injury had increased inflammation and demyelination that was not significant from each other. The inconsistent findings with SIRT1 and NRF2 make future clinical applications difficult to consider. Previous findings have indicated that NRF2 preserved RGCs following ON crush but did not promote axonal regeneration, which is required for its function and may help explain why NRF2 was able to significantly increase RGC counts but had no improvement on OKR [54]. In addition, the group reported a lackluster RGC transduction of only 21% with their vectors [19].

Another group investigated an AAV2-NRF2 encoding vector with a selective monocyte chemoattractant protein 1 (Mcp-1) promoter (AAV2-pMcp-1-NRF2) on an ON crush model in mice [20]. Mcp-1 is involved in leukocyte recruitment and capable of driving gene expression in stressed RGCs preceding cell death [20,55]. Therefore, an aspect of selectivity and AAV2 gene vector expression regulation was considered by utilizing a promoter more selective for injured RGCs. NRF2 expression was significantly increased in injured mice compared to non-injured mice. Treatment with AAV2-pMcp-1-NRF2 on ON crush mice also significantly reduced the number of Sytox positive cells, indicating RGC death, compared to untreated mice. The use of the Mcp-1 or a CMV promoter had no significant difference in the number of Sytox^+^ cells or Brn3a^+^ RGCs. Visual acuity and contrast sensitivity was also not significantly different between the different promoters. However, AAV2-pCMV-NRF2 treatment significantly increased the gene expression of *Ho-1*, *Atf4*, and *p53*. *Ho-1* is a direct transcription target of NRF2, while *Atf4* is an endoplasmic reticulum stress marker, and *p53* is involved in the cell death pathway [20]. The use of a selective marker that targets stressed RGCs potentially addresses off-target and unknown long-term effects on healthy RGCs and elucidates a possible mechanism to regulate AAV vector gene therapy.

### 2.5. AAV Vectors Encoding Other Genes Therapy Approaches

Exoenzyme C3 transferase (C3) can inactivate Rho GTPases, which are involved in cellular processes such as the regulation of cell proliferation and apoptosis [21]. C3 can protect RGCs from NMDA induced excitotoxicity in rats [56]. One group evaluated a self-complementary AAV encoding C3 (scAAV-C3) in a I/R injury model in rats [21]. The I/R was induced through the temporary elevation of IOP to 110 mmHg for 90 min with saline. This injury model does not accurately emulate the chronic and progressive pathophysiology of glaucoma. The rats were also injected one week prior to induction of I/R injury. Injured rats treated with scAAV-C3 had significantly less TUNEL positive and cleaved caspase-3 positive cells in the retina compared to both injury only and injury + GFP rats, indicating attenuated apoptosis. There was also a partial rescue effect of retinal neurons and RGCs, as injured + treated rats had a significantly higher number of retinal neurons and RGCs compared to injury only and injury + GFP rats. Injured + treated rats had a significantly attenuated loss of retinal thickness compared to injury only and injury + GFP rats.

Deletion of DNA-damage inducible transcript 3 (*Ddit3*) and sterile α and TIR motif-containing protein 1 (*Sarm1*) can preserve the structure and function of RGC soma and axons [57,58]. *Ddit3* encodes CCAAT/enhancer binding protein homologous protein (CHOP), which has been shown to be upregulated in optic nerve injury [58]. *Sarm1* is a prodegenerative signal in the axon death signal pathway [57]. One group encoded an AAV2 encoding a mouse γ-synuclein (AAV2-mSncg) promoter-driven CRISPR/Cas9 vector with *Ddit3* + *Sarm1* guide RNA (gRNA) and assessed the effects on an ON crush injury model in mice [22]. The promoter, mSncg, was chosen because it was the most specific for mouse RGCs compared to other commonly used promoters to drive gene expression in neurons. Interestingly, mSncg targeted human pluripotent stem cell (hPSC)-derived human RGCs more effectively than human Sncg. AAV2-mSncg promoter-driven CRISPR/Cas9 remained expressed in mice RGCs for up to 12 months post injection. Treatment with AAV2-mSncg promoter-driven CRISPR/Cas9 and *Ddit3* + *Sarm1* gRNA was able to decrease the expression of both Ddit and *Sarm1*. There was a significant reduction in *Ddit3* positive RGCs, although *Sarm1* was not evaluated similarly. Treatment with AAV2-mSncg promoter-driven CRISPR/Cas9 and *Ddit3* + *Sarm1* gRNA in ON crush was able to significantly protect ganglion cell complex thickness, RGC somata, and RGC axons compared to controls. The model for injury was an acute ON crush model and mice were injected with treatment two weeks prior to the induction of the model. Although the CRISPR/Cas9 system is designed to be highly specific, it did not drive gene expression exclusively in RGCs. In combination with the long-term expression, further studies are required to fully elucidate any long-term effects for clinical usage.

Pigment epithelium-derived factor (PEDF) is a member of the serpin super-family of serine protease inhibitors that exhibit antiangiogenic, neurotrophic, and neuroprotective activities, and in combination with mesenchymal stem cells (MSCs), PEDF can provide trophic support through the release of neurotrophic factors and protect central neurons after ON crush [59,60]. One group studied the effects of AAV2 encoding PEDF (AAV2-PEDF) in combination with human MSCs (hMSCs) to determine whether a synergistic effect on RGC survival and axonal growth could be established in an ON crush injury rat model [23]. AAV2-PEDF treatment in injured rats was able to significantly increase the number of RGCs compared to GFP only rats. PEDF treatment also significantly increased the number of neurotrophic factor, FGF-2, immunostained cells and significantly decreased the number of IL-1β immunostained cells compared to GFP only. PEDF treatment was able to significantly decrease Iba1^+^ cells and Müller GFAP^+^ cells compared to GFP only, suggesting a decrease in the microglial response and Müller glia activation. PEDF treatment alone was not able to promote axonal outgrowth, but in combination with hMSCs, it was able to significantly increase CTB^+^ axons, which is a marker for the anterograde tracing of RGC axons. However, this significant increase was only marked at 0.25 mm away from the injury site, and all measured distances further than 0.25 mm were not significantly different from GFP only. The intravitreal injection of PEDF occurred four weeks prior to ON crush, and hMSCs were injected immediately after ON crush. Therefore, with this specific strategy, the temporal timing of two separate injections would have to be considered in clinical settings.

ON and retinal damage in glaucoma is preceded by local complement upregulation and activation [24]. Complement C3 activation intersects the three pathways of complement activation, and one group assessed an AAV vector encoding a C3 inhibitor, CR2-Crry (AAV2-CR2-Crry), in the DBA/2J mouse model of glaucoma [24]. IOP increases in DBA/2J mice from 7 months to 9 months naturally in these mice, but treatment with AAV2-CR2-Crry did not lower IOP. Mice treated with AAV2-CR2-Crry had significant differences in the density of Brn3a^+^ cells in the retina compared to control and GFP only mice. However, these differences were categorized into pre-degenerative, declining, and degenerative cells, and not quantified separately. Treatment with AAV2-CR2-Crry was also able to significantly increase axon fasciculations compared to control and GFP only mice. AAV2-CR2-Crry treatment was able to attenuate ON pathology compared to control and GFP only mice. However, this was assessed qualitatively with sections under light microscopy without using an established marker of ON pathology. These mice were injected with treatment or GFP at 7 months of age and assessed at 10 and 12 months for neurodegeneration. As mentioned before, complement upregulation precedes ON and retinal damage in glaucoma. Therefore, the timing of inhibiting C3 could make potential therapeutic use difficult to manage as most glaucoma is asymptomatic until a relatively late stage [3].

Stanniocalcin-1 (STC-1) is a protein that is protective in a variety of tissues, including RGCs [61]. Although the exact mechanisms are unclear, STC-1 has been shown to be physiologically involved in the regulation of cellular calcium and phosphate homeostasis [61]. One group studied the effects of an AAV vector encoding STC-1 (AAV-STC-1) in two transgenic mice models of retinal degeneration (P23H-1 and S334ter-4) [25]. Both transgenic mice treated with AAV-STC-1 had significantly thicker optic nerve layers compared to untreated mice, but only P23H-1 treated mice had significantly improved scotopic and photopic B-waves when ERG responses were assessed. Treatment with AAV-STC-1 caused changes in gene expression in both transgenic mice, but there were a greater number of genes both up- and downregulated in S334ter-4 mice than in P23H-1 mice. Some of the genes that were significantly upregulated in S334ter-4 mice included *Gpnmb*, *Hmox*, *Hspb1* (Hsp27), *Nupr*, *Serpinf1* (PEDF), *UCP*-2, and *USP18*, which are all either neuroprotective, anti-oxidative/anti-reactive oxygen species, or anti-apoptotic. Interestingly, these genes were not significantly upregulated in P23H-1 mice. Testing AAV-STC-1 in two separate transgenic models of retinal degeneration helps to provide further evidence of treatment effectiveness. However, the inconsistencies in results and induction of gene expression of STC-1 (38 ± 16-fold induction in P23H-1 and 292 ± 107-fold induction in S334ter-4 eyes) indicates the need for further studies to elucidate the effectiveness and safety for future clinical applications.

Neuroglobin (NGB) is a globin with high affinity for oxygen that is preferentially expressed in neurons and is thought to be involved in neuronal protection from hypoxia or oxidative stress [62,63]. As a powerful neuroprotectant, knockdown of the *Ngb* gene in rats caused RGC death, optic neuropathy, and visual functional impairment [62,63]. One group investigated an AAV vector encoding NGB (AAV2-NGB) in the DBA/2J mouse model of glaucoma [26]. Mice were treated either early (2 months) or late (8 months) and RGC counts were only significantly increased in mice treated early compared to untreated controls. However, retinal stress as measured with GFAP showed that both early and late treatment were able to significantly reduce GFAP fluorescence intensity compared to untreated control mice. Morphological changes in the ganglion cell layer were also observed. Early treatment resulted in RGC morphology and dendritic branches comparable to early age untreated mice, before the onset of disease progression. Late treatment led to better preserved RGC morphology and dendritic branches compared to age-matched untreated mice. However, these morphological changes were qualitative assessments. As RGC count improvement required early treatment, and early glaucoma being relatively asymptomatic, the temporal factor would make NGB a difficult gene therapy strategy to use in clinical applications.

Fas ligand (FasL) plays an important role in homeostasis and the self-tolerance of lymphocytes. It can be expressed either as a membrane-bound protein (mFasL), which is pro-apoptotic and pro-inflammatory, or cleaved in a soluble form (sFasL), which is antagonist and inhibits the activity of mFasL [64]. One group explored an AAV2 vector encoding sFasL (AAV2-sFasL) on the DBA/2J mouse model of glaucoma and microbead-induced elevated IOP in mice [27]. Intravitreal injection of AAV2-sFasL was given at 2 months of age and significantly increased RGC density and axon density compared to GFP only at 10 months and 15 months. Treatment with AAV2-sFasL also significantly reduced mRNA expression of the pro-apoptotic mediators Fas, FADD, and BAX, and significantly increased the anti-apoptotic mediators cFLIP, Bcl2, and cIAP2. Qualitative confocal microscopy images revealed reduced Müller glial cells, along with a significant reduction of GFAP and TNFα mRNA expression, suggestive of decreased Müller glial cell activation. The effects of AAV2-sFasL treatment on microbead-induced elevated IOP in mice was also assessed, and RGC and axon density were significantly increased compared to GFP only. The injection of AAV2-sFasL was given three weeks prior to the microbead induction of elevated IOP. Although this group was able to demonstrate effectiveness in two separate mouse models of injury, the mechanism behind how sFasL attenuates mFasL remains to be determined, and the appropriate balance between the membrane bound and cleaved form remain unknown.

### 2.6. MicroRNA Gene Based Therapy Approaches

MicroRNA-21 (miR-21) is one of the most common and highly upregulated miRs that negatively module gene expression. When overexpressed in cultured astrocytes, miR-21 leads to decreased cell size, process thickness, and GFAP expression [28,65]. One group explored the effects of miR-21 on ON crush in rats [28]. Intravitreal injections of miR-21 mimic (agomir) and miR-21 inhibitor (antagomir) were given immediately after ON crush. Injured + antagomir rats had significantly more axons compared to the negative control. F-VEP amplitude was significantly higher in injury + antagomir treated rats compared to injury + agomir rats. There was no significant difference in either antagomir or agomir treated rats in F-VEP latency. Injury + antagomir rats had significantly increased RGCs compared to injury only. The group was able to elucidate the possible mechanism of how the inhibition of miR-21 can induce an environment more conducive to axonal regeneration and functional recovery of ON injury. Inhibition of miR-21 decreased protein levels of EGFR/PI3K/AKT/mTOR, thereby attenuating excessive astrocyte activation and glial scar progression, allowing for axon regeneration and functional recovery after ON crush. Excessive astrocyte activation and glial scar formation is detrimental to axon regeneration after ON injury.

## 3. Discussion

BDNF is a neurotrophic factor that has functions both within and outside of the central nervous system (CNS) [29]. BDNF is able to regulate the survival, development, function, and plasticity of neurons [29]. BDNF has also been demonstrated to provide a neuroprotective effect under injurious conditions, such as glutamatergic stimulation, cerebral ischemia, hypoglycemia, and neurotoxicity [66]. RGCs extend their axons through the optic nerve to the brain; as such, they are considered part of the CNS, and the neuroprotective effects of BDNF are also demonstrated on RGCs [29]. BDNF is able to exert its neuroprotective functions by binding to its receptor, tropomyosin receptor kinase B (TrkB) [29]. There is increasing evidence that intravitreal injection of BDNF can alleviate RGC loss [67,68,69]. Increasing evidence of reduced BDNF and TrkB signaling in human glaucoma has also been suggested [70,71,72,73]. Therefore, BDNF’s well established neuroprotective role, increasing evidence for neuroprotection in RGCs, and its potential pathophysiological role in human glaucoma most likely explain the higher proportion of papers exploring novel gene therapy approaches with BDNF.

Aside from erythropoietin’s (EPO) role in red blood cell production, it has also been shown to exert neuroprotective effects in models in which neurons are lost [44,45]. In a clinical trial study with human patients with acute ischemic stroke, EPO was shown to significantly reduce infarct size and improve functional outcomes when it was administered eight hours after stroke onset [46]. EPO has historical usage in anemic patients with renal failure and is thus considered safe to use in humans [45]. EPO is able to exert this neuroprotective effect through the activation of Janus-tyrosine kinase (JAK)-2, which is linked to the EPO receptor [45]. JAK-2 then phosphorylates many downstream signaling factors, including signal transducers and activators of transcription (STAT)-5, which stimulates the mitochondria to release anti-apoptotic proteins such as B-cell lymphoma-extra large (Bcl-xL) [45]. Bcl-xL goes on to inhibit cytochrome *c*-dependent caspases, which ultimately prevents apoptosis [45]. These neuroprotective effects were extended and observed in light-induced retinal degeneration in retinal ischemia and RGCs [44,74,75]. EPO’s established clinical use and the increasing evidence of its neuroprotective effect may explain why two of the 18 studies focused on EPO.

NF-E2-releated factor 2 (NRF2) is a transcription factor that has a key role in the systemic antioxidant defense system [76]. NRF2 initiates the transcription of a cascade of antioxidant enzymes and its potential protective role through attenuated oxidative stress has been demonstrated in diabetic retinopathy, retinal I/R injury, and optic neuritis [77,78,79,80]. Increasing evidence has also demonstrated that oxidative stress can lead to the loss of RGCs in many ocular neurodegenerative diseases. Therefore, combined with NRF2’s role in antioxidant defense, this may explain its focus as a target for gene therapy.

Most gene therapy strategies (17/18 studies) utilized an adeno-associated-viral vector (AAV). AAV vectors are the leading platform for the in vivo delivery of gene therapies [81]. In 2017, LUXTURNA™ became one of the few AAV vector based gene therapies approved by the US Food and Drug Administration to treat Leber’s congenital amaurosis [13]. Promising therapeutic potential with an AAV focused vector may explain why all but one study utilized this system for gene therapy. Despite the potential therapeutic evidence for BDNF, EPO, NRF2, C3, Scng guided CRISPR/Cas9, PEDF + hMSC, CR2-Crry, STC-1, NGB, sFasL, and miR-21, many shared limitations in all the studies included a limited model of glaucoma or related ocular disease, pre-emptive intravitreal injection of the gene therapy, and a lack of regulation of gene therapy expression.

Utilizing an AAV vector allows for unregulated expression of the therapeutic gene in AAV-transduced cells, which includes healthy and injured cells of interest. Therefore, the unregulated expression of a therapeutic gene in healthy cells could potentially result in stress and a non-physiological condition in otherwise previously healthy cells. Therefore, the proper regulation of these AAV vectors is paramount for their safety and effectiveness before clinical application can be considered. One of the studies tried to address this regulation by utilizing a specific promoter that is activated during RGC stress [20]. Another study utilized a CRISPR/Cas9 system with a guide RNA specifically targeted to RGCs [22]. Although these two studies provide novel ways to address the specific targeting of relevant RGCs, once transduced into target cells, there is no way to regulate the expression of the therapeutic gene. As one study also mentioned, despite potential therapeutic effects, it is important to be able to address the appropriate balance of therapeutic genes under normal and therapeutic conditions [27]. In other words, being able to appropriately regulate gene expression after AAV transduction in cells would confer more confidence in the safety and effectiveness of these types of gene therapy.

The use of mRNA has garnered attention as a safer alternative to plasmids, viral vectors, and artificial chromosomes for gene therapy [82]. With mRNA gene therapy, there is regulated expression through its inherent transient expression, the lack of nucleus entry and therefore the lack of insertional mutagenesis [82,83]. Currently, with the COVID-19 pandemic, mRNA vaccines have marked a critical milestone for the usage of mRNA based therapy [84]. Another RNA based gene therapy strategy is the use of miRNAs that bind and inhibit complementary mRNA targets [85]. The inappropriate expression of miRNA has been linked to a variety of diseases, and there have been many studies utilizing miRNA to treat neurodegenerative diseases [86]. However, unlike mRNA therapy, miRNA have not reached a critical milestone in clinical use, and its safety and efficacy needs to be ensured as overexpression has caused organ failure and death in rats [85,87].

Despite the increasing evidence for gene therapies as potential future clinical solutions to address ocular diseases, including glaucoma, further evidence to study long-term safety, effectiveness, and therapeutic gene regulation are required.

## 4. Conclusions

Based on a systematically approached literature review, gene-based therapy for RGC neuroprotection has promising clinical applications, as shown by the results in both in vivo and in vitro models. However, there is still much work to be done before such strategies can move forward with clinical trials. Namely, the models of injury in many of the studies were more acute in nature, unlike the more progressive and neurodegenerative pathophysiology of diseases such as glaucoma. The regulation of gene expression is also highly unexplored, despite the use of AAV vectors in the majority of the studies reviewed. Therefore, IOP-independent strategies, such as gene-based therapy, have the potential to prevent and improve neurodegenerative disease progression, but there are still many more steps that need to be considered before moving to clinical trials.

## Figures and Tables

**Figure 1 biomolecules-11-00581-f001:**
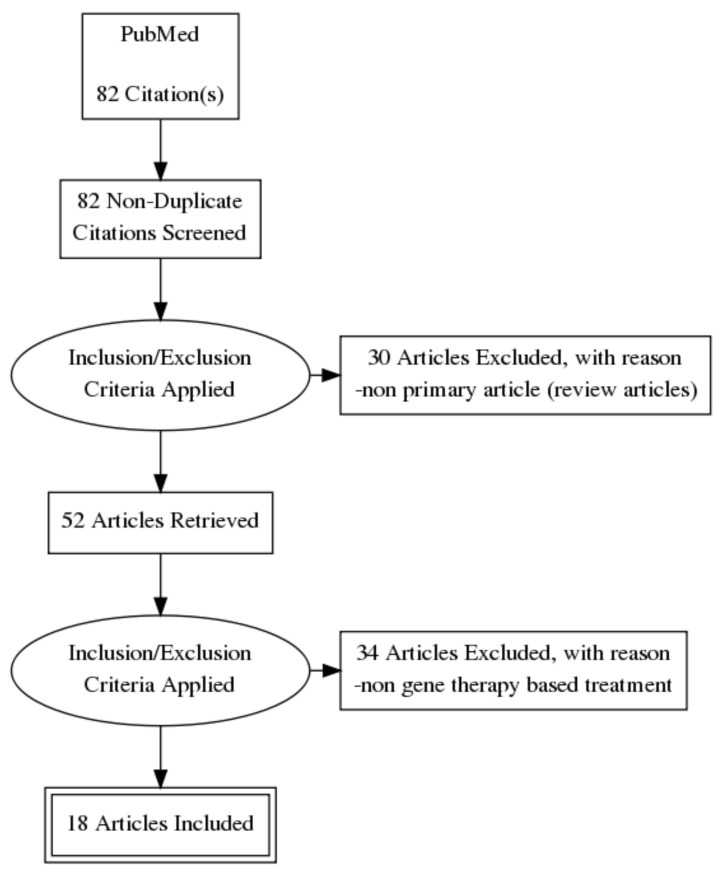
Preferred reporting items for systematic reviews and meta-analyses (PRISMA) flowchart illustrating the selection process of the articles.

**Table 1 biomolecules-11-00581-t001:** Summary of in vivo and in vitro studies utilizing an adeno-associated viral (AAV) vector encoding brain-derived neurotrophic factor (BDNF) as gene therapy.

Reference/Study	Country	Injury Model and Animal	Groups and Sample Size	Parameters for Efficacy	Main Result	Remarks
**1. Viral vector encoding brain-derived neurotrophic factor (BDNF)**						
a. Adeno-associated viral vector gene therapy (AAV2) encoding BDNF (AAV2-BDNF) Wójcik-Gryciuk A. et al., 2020 [11]	Poland	Microbead induced elevated IOP, rat	12 negative control, 14 damaged only, and 13 damage + treatment	RGC count BDNF and TrkB protein levels	Significant attenuation of RGC loss in AAV2-BDNF treated rats (1267 ± 60 RGCs/mm^2^) vs. glaucoma induced rats (758 ± 124 RGCs/mm^2^). Unilateral injection of BDNF, bilaterally upregulated BDNF levels and restoration of TrkB levels back to normal levels.	Neuroprotection of RGCs indicated by the attenuated loss of RGCs. Unilateral injection of BDNF was able to upregulate BDNF in both injected and non-injected eyes.
b. AAV2 gene therapy encoding BDNF and tropomyosin related receptor kinase-B (AAV2 TrkB-2A-mBDNF) Osborne A. et al., 2018 [12]	United Kingdom	**In vitro** oxidative damage via hydrogen peroxide, immature SH-SY5Y human neuroblastoma cells	4 negative control, 4 empty vector control, 4 TrkB + BDNF treated, 4 TrkB treated only, 4 BDNF treated only	Neuroprotective effect via TUNEL measurement	TrkB + BDNF treated SH-SY5Y cells significantly reduced TUNEL positive apoptotic cells.	The neuroprotective effects of just BDNF or TrkB treatment alone were not assessed when damaged with hydrogen peroxide.
c. AAV2 gene therapy encoding BDNF and TrkB (AAV2 TrkB-2A-mBDNF) Osborne A. et al., 2018 [13]	United Kingdom	Optic nerve crush, mice and rats	Cohort 1: 16 positive control Cohort 2: 12 TrkB + BDNF treated mice Cohort 3: 36 TrkB + BDNF treated mice Cohort 4: 25 BDNF treated mice Cohort 5: 22 TrkB + BDNF treated mice study mice Cohort 6: 15 TrkB + BDNF treated mice and 15 BDNF treated only Cohort 7: 20 TrkB + BDNF treated mice and 20 TrkB treated only Cohort 8: 30 TrkB + BDNF treated mice	Brn3a positive RGC count Glial fibrillary acidic protein (GFAP) activation Visual function via electroretinography (ERG) Axon and RGC counts in rats with laser-induced ocular hypertension (OHT)	BDNF treatment significantly rescued Brn3a^+^ RGCs. TrkB + BDNF treatment significantly rescued Brn3a^+^ RGCs and there was a significantly greater rescue effect with both TrkB + BDNF compared to BDNF treatment alone. No significant difference in GFAP levels in TrkB + BDNF groups compared to GFP control group and negative control up to 24 weeks. Significant improvement in positive scotopic threshold response (pSTR) in TrkB + BDNF treated mice in optic nerve crush compared to just TrkB alone. No significant changes in negative A- or positive B-wave responses in any of the treatment groups. TrkB and BDNF were able to significantly protect and restore axon counts and RGCs in rats.	TrkB + BDNF were able to significantly rescue more RGCs than BDNF treatment only No adverse effects on retinal health with TrkB + BDNF treatment as suggested by a lack of change in GFAP levels. Only TrkB + BDNF and TrkB treatment were compared with BDNF treatment only. Only TrkB + BDNF and TrkB treatment only were compared with BDNF treatment only.
d. Tyrosine triple mutated AAV2-BDNF (tm-scAAV2-BDNF) gene therapy Shiozawa A.L. et al., 2020 [14]	Japan	*N*-methyl-D-aspartate (NMDA) induced retinal injury, mouse	6 negative control, 6 damage, 9 damaged + treatment	Inner retinal layer thickness Brn3a positive RGC count Visual function via scotopic ERG GFAP activation	Significant attenuation of inner retinal thickness loss in damage + treatment mice vs. damage only mice. Significant prevention of Brn3a^+^ RGC loss in damage + treatment mice vs damage only mice. Significantly higher A- and B-wave amplitudes in the damage + treatment mice vs. damage only mice. Significant reduction in GFAP in the damage + treatment mice vs. damage only mice.	Protection and prevention of histological changes caused by NMDA. Neuroprotection of RGCs indicated by an attenuated loss of RGCs. Suggested preservation of visual function. Reduction in retinal damage using GFAP as a maker for retinal stress.
e. tm-scAAV2-BDNF gene therapy Igarashi T. et al., 2016 [15]	Japan	Increased IOP via saline infusion to induce retinal ischemia/reperfusion (I/R), mouse	6 control, 6 damage, 6 damage + empty vector, 6 damage + treatment	Inner retinal layer thickness Brn3a positive RGC count Visual function via scotopic ERG GFAP activation	Significant attenuation of inner retinal thickness loss in damage + treatment mice (45.4 ± 4.2 µm) vs. damage + empty vector mice (30.2 ± 3.0 µm). Significant prevention of Brn3a^+^ RGC loss in damage + treatment mice (22.6 ± 0.5 cells) vs. damage + empty vector mice (11.2 ± 0.8 cells). Significantly higher B-wave amplitudes in damage + treatment mice (862.6 ± 146.6 µv) vs. damage + empty vector mice (484.8 ± 201.6 µv). Significant reduction in GFAP in damage + treatment mice (4.8 ± 4.4) vs. damage + empty vector mice (12.3 ± 0.6)	Protective effects of BDNF indicated by the thicker retinal structure and no significant difference with normal mice. Neuroprotection of RGCs indicated by an attenuated loss of RGCs. Suggested preservation of visual function. Reduction in retinal damage using GFAP as a maker for retinal stress.
f. AAV2-BDNF gene therapy and/or an AAV2 vector encoding a mutated phosphor-resistant version of collapsing response mediator protein 2 (AAV2-CRMP2T555A) Chiha W. et al., 2020 [16]	Australia	Unilateral partial transection of the optic nerve, rats	11 negative control, 11 control vector with GFP, 6 BDNF + GFP, 9 BDNF only, 9 CRMP2T555A only, 10 BDNF + CRMP2T555A	Visual function via optokinetic nystagmus response RGC count Axonal density in the optic nerve Structural disruptions at the node of Ranvier complex Myelin thickness via G-ratio Oxidative stress and inflammatory cell markers	Injured rats with BDNF (BDNF only and BDNF + CRMPT2T555A) had a significantly increased total number of pursuits compared to injured rats with control vector with GFP injections. In the central retina, BDNF, only, CRMPT2T555A only, and BDNF + CRMPT2T555A significantly increased RGC counts compared to injured control vector groups in the central retina when counting βIII-tubulin^+^ RGCs. When counting Brn3a^+^ RGCs, only CRMP2T555A only and BDNF + CRMP2T555A significantly increased RGC counts. In the ventral retina, BDNF only significantly increased RGC counts with either βIII-tubulin^+^ or Brn3a^+^ RGCs. BDNF only, CRMP2T555A only, and BNDF + CRMP2T555A treatment significantly restored the number of optic nerve axons. BNDF + CRMP2T555A treatment significantly restored the paranodal gap when defined as two Caspr^+^ paranodes. BDNF only and BNDF + CRMP2T555A treatment significantly restored the paranodal gap when defined as βIII-tubulin^+^ axons flanked by Caspr^+^ paranodes. BDNF only, CRMP2T555A only, and BNDF + CRMP2T555A treatment significantly reduced the G-ratio and myelin thickness. Only BNDF + CRMP2T555A treatment was able to significantly reduce HNE, an oxidative stress marker, while there was no affect from any treatment groups on the microglia marker, Iba1.	Partial ON transection has been previously reported to reduce the optokinetic nystagmus response. Total pursuits included smooth, partial, and micro pursuits. Smooth and fast rests were not significantly different. βIII-tubulin^+^ RGCs were also counted as not all RGC subtypes express Brn3a, and RGC loss after injury is often preceded by a downregulation in Brn3a. For axonal density, βIII-tubulin^+^ RGCs were also counted as not all RGC subtypes express Brn3a, and RGC loss after injury is often preceded by a downregulation in Brn3a. Paranodal gaps defined as βIII-tubulin^+^ axons flanked by Caspr^+^ paranodes represents a more rigorous definition for the Node of Ranvier. G-ratio is the ratio of the inner axonal diameter to the total diameter and it is used to assess axonal myelination. HNE increases in ONs vulnerable to secondary degeneration.

**Table 2 biomolecules-11-00581-t002:** Summary of in vivo studies utilizing an AAV vector encoding erythropoietin (EPO) as gene therapy.

Reference/Study	Country	Injury Model and Animal	Groups and Sample Size	Parameters for Efficacy	Main Result	Remarks
**2. Viral vector encoding erythropoietin (EPO)**						
a. AAV2-EPO gene therapy Tao Y. et al., 2020 [17]	China	*N*-methyl-*N*-nitrosourea (MNU) induced retinal degeneration, mice	10 negative control, 10 damage + treatment, 10 damage only, 10 treatment only	Visual function via ERG responses Behavioural function via vision-guided optokinetic tests Retinal layer thickness Cone photoreceptor count Multi-electrode array (MEA) field potentials RGC spontaneous firing spikes Protective mechanism	Scotopic B-wave amplitudes and photopic ERG responses in the damage + treatment group were significantly higher than the damage only group. Visual acuity and contrast sensitivity was significantly higher in the damage + treatment group than in the damage only group. Retina thickness was significantly larger in the damage + treatment group compared to the damage only group. The number of peanut agglutinin (PNA)^+^ cone photoreceptors was significantly higher in the damage + treatment group compared to the damage only group. The mean amplitude of field potential was significantly larger in the damage + treatment group compared to the damage only group. The spontaneous firing rate of RGCs in the damage + treatment group was significantly lower compared to the damage only group. The mRNA levels of caspase-3, CHOP, and Bax were significantly downregulated in damage + treatment group vs. damage only. The mRNA level of Bcl-2 was significantly upregulated in damage + treatment group vs. damage only.	The damage + treatment group was not significantly different than the control group. Visual acuity and contrast sensitivity in the damage + treatment group were not significantly different from the control group. The damage + treatment group was not significantly different than the control group. PNA staining was absent in the damage only group. Field responses in the damage + treatment group were not homogenous. Peripheral regions were significantly higher than mid-peripheral and central regions. Mid-peripheral regions were significantly larger than central regions. The damage + treatment group still had significantly higher spontaneous firing rates compared to normal mice.
b. rAAV.EpoR76E gene therapy Hines-Beard J. et al., 2016 [18]	United States	DBA/2J mouse model of glaucoma	15 negative control, 25 GFP vector control, 30 treated mice	Protection against vision loss Microglia number Microglia morphology Pro-inflammatory cytokines/chemokines Antioxidant enzymes	The N1 and P1 amplitudes in treated mice were not significantly different from negative control or GFP vector control mice. The number of microglia were significantly reduced in treated mice vs GFP vector control mice in both central and peripheral retina. The average soma area of treated mice was significantly smaller compared to GFP vector control mice. Treated mice had significantly reduced IL-1α, IL-1β, IL-17, IL-12p40, IL-12p70, CCL4 (MIP-1β), CCL5 (RANTES), and IL-13 mRNA levels compared to GFP vector control mice. There was an increase in the expression of several antioxidant proteins in both GFP vector control and treated mice compared to negative control.	IOP was not significantly different between treated and GFP vector control mice. Microglia numbers increase prior to the onset of IOP elevation in this mouse model. The average soma area of treated mice was significantly different than negative control mice There was no significant difference in TNFα levels between treated and vector control mice. These increases were not significant.

**Table 3 biomolecules-11-00581-t003:** Summary of in vivo studies utilizing an AAV vector encoding nuclear factor erythroid 2-related factor 2 (NRF2) as gene therapy.

Reference/Study	Country	Injury Model and Animal	Groups and Sample Size	Parameters for Efficacy	Main Result	Remarks
**3. Viral vector encoding NRF2**						
a. AAV2-NRF2 and AAV2-Sirtuin 1 (SIRT1) gene therapy McDougald DS. et al., 2018 [19]	United States	Induction of experimental autoimmune encephalomyelitis (EAE), mice	10 negative control, 10 injury + vehicle, 10 GFP only, 10 injury + GFP, 25 AAV2-NRF2, 25 AAV2-SIRT1	Visual acuity RGC count via Brn3a Immune cell infiltration via H&E staining and demyelination measured via LFB staining	Injured mice treated with AAV2-SIRT1 had significant improvement in optokinetic responses (OKR) compared to injury + GFP mice. Injured mice treated with AAV2-NRF2 had significantly higher numbers of Brn3a^+^ RGCs compared to injury + vehicle. AAV2-SIRT1 treatment did not significantly increase RGCs compared to injury + vehicle mice. Both NRF2 and SIRT1 had no significant impact or attenuation of immune cell infiltration or demyelination of ON sections.	There was no significant improvement in OKR with AAV2-NRF2 treatment. There was no significant difference with NRF2 treatment compared to injury + GFP, despite any purposeful therapeutic gene encoded in the AAV2-GFP vector. There was no significant difference among all groups that were injured, indicating no protective or therapeutic response of NRF2 or SIRT1 treatment.
b. AAV2-pMcp-NRF2 gene therapy Fujita K. et al., 2017 [20]	Japan	ON crush model, mice	8 negative control, 8 pMcp promoter treatment, 8 CMV promoter treatment	mRNA expression of target gene (NRF2) RGC death measured via Sytox Orange RGC count measured via Brn3a, Brn3b, and Thy1 Mcp1 promoter and CMV promoter driven AAV2-NRF2 treatment Gene expression differences in Mcp1 and CMV promoter driven AAV2-NRF2 treatment	Mice with ON crush had significantly higher expression of NRF2 compared to mice treated with AAV2-pMcp-NRF2 without injury. Treated mice had significantly reduced numbers of Sytox positive cells in ON crush eyes. RGC makers (Brn3a, Brn3b, and Thy1) were significantly increased with NRF2 treatment compared to non-injured controls. Treatment with Mcp1 promoter or CMV promoter driven AAV2-NRF2 conferred no differences in mRNA expression of RGC markers or visual acuity and contrast sensitivity. CMV promoter driven AAV2-NRF2 significantly increased *Ho-1*, *Atf4*, and *p53* compared to control and Mcp-1 promoter driven AAV2-NRF2.	An increase in NRF2 gene expression selectively in injured eyes suggests positive selectivity with the pMcp promoter. Sytox Orange is a nucleic acid stain for dead cells. Relative mRNA levels were presented, not RGC counts. Despite a Mcp1 promoter being utilized to be more specific for stressed RGCs, the effects seem to be no different from a CMV promoter. *Ho-1* is a direct transcription target of NRF2, *Aft4* is an ER stress marker, and *p53* is involved in the cell death pathway.

**Table 4 biomolecules-11-00581-t004:** Summary of in vivo studies utilizing an AAV vector encoding different genes as gene therapy.

Reference/Study	Country	Injury Model and Animal	Groups and Sample Size	Parameters for Efficacy	Main Result	Remarks
**4. Viral vector encoding other genes**						
a. scAAV2-C3 gene therapy Tan J. et al., 2020 [21]	China	I/R injury induced by elevated IOP by saline injection, rats	4 negative control, 4 injury only, 4 GFP only, 4 injury + GFP, 4 treatment only, 4 injury + treatment	RhoA inhibition Apoptosis measured via TUNEL positive and cleaved caspase-3 positive cells Retinal neuron and RGC counts via NeuN and Brn3a staining Retinal thickness	Injury + treated rats had significantly lower protein levels of RhoA compared to negative control and GFP only rats Injury + treated rats had significantly lower TUNEL positive cells compared to injury + GFP rats and injury only rats. Injury only and injury + GFP groups had a significantly higher number of caspase-3 positive cells compared to negative control, and injury + treated rats had significantly reduced caspase-3 positive cells compared to injury only and injury + GFP groups. NeuN positive cells were significantly rescued in the injury + treatment group compared to both injury only and injury + GFP rats. Brn3a positive cells were significantly rescued in the injury + treatment group compared to injury + GFP rats. Injury + treated mice were able to significantly attenuate retinal thickness loss compared to injury only and injury + GFP rats.	RhoA has involvement in apoptosis and C3 has been shown to inhibit Rho and protect RGCs from NMDA induced damage. There was an absence of TUNEL positive cells in the groups with no injury. Injury only and negative control rats were not used in the Brn3a experiments. There was still a significant difference between injury + treated rats and the negative control and GFP only rats, indicating only a partial attenuation.
b. AAV-mSncg promoter-driven CRISPR/Cas9 gene therapy Wang Q. et al., 2020 [22]	United States	ON crush injury model, mice	8 control gRNA in LSL-Flag-SpCas9 mice, 9 target gRNA in LSL-Flag-SpCas9 mice, 8 control gRNA in WT mice, 9 target gRNA in WT mice	AAV-mSncg promoter and GFP expression *Ddit3* and *Sarm1* expression Ganglion cell complex (GCC) thickness, RGC somata, and RGC axons	The mouse Scng promoter (AAV-mSncg-EGFP) binds and targets hPSC-derived human RGCs expressing tdTomato based on merge labelling (of GFP and tdTomato) significantly more than the human Scng promoter (AAV-hSncg-EGFP). Mice treated with AAV-mSncg-CRISPR/Cas9 with *Ddit3* and *Sarm1* guide RNA had reduced expression of Ddit3-mCherry and Sarm1-mCherry. Mice treated with AAV-mSncg-CRISPR/Cas9 with *Ddit3* and *Sarm1* guide RNA after ON injury had significantly thicker GCC, increased RGC somata, and RGC axons compared to non-treated mice.	Yellow fluorescence was indicated as proper targeting of inherent tdTomato expression in the hPSC-derived human RGCs and appropriate AAV-mSncg-EGFP targeting. Only the expression of *Ddit3* mCherry was significantly decreased with AAV-mSncg-CRISPR/Cas9 with *Ddit3* and *Sarm1* guide RNA treatment. *Sarm1* mCherry expression was not assessed in the same way. There were significantly more RGC somata in treated WT mice compared to treated LSL-Cas9 mice with ON injury.
c. AAV2-pigment epithelium-derived factor (PEDF) + human mesenchymal stem cell (hMSC) gene therapy **Nascimento-dos-Santos G. et al., 2020 [23]**	Brazil	ON crush injury model, rats	9 control, 5 GFP only, 7 PEDF only, 9 PEDF + hMSC, 3 GFP + hMSC	RGC neuroprotection assessed via Tuj1^+^ cells Neuroprotective factor FGF-2 and IL-1β Microglial response and Müller glia activation Axonal outgrowth	Injured mice that were treated with PEDF had a significantly higher number of Tuj1^+^ cells compared to GFP only. Injured mice treated with PEDF had significantly higher levels of FGF-2 and significantly lower IL-1β compared to GFP only. Injured mice treated with PEDF had a significantly lower number of Iba1^+^ cells and Müller GFAP^+^ cells compared to GFP only. PEDF treatment after injury did not stimulate RGC axonal regrowth. However, the combination of PEDF + hMSC was able to significantly increase the number of Tuj^+^ cells and CTB^+^ axons.	Significance was not reported between PEDF treated and control mice, although it seems that PEDF treatment was not able to increase Tuj^+^ cells to control levels. There was no significant difference between control and PEDF treatment groups, suggesting levels comparable to normal state. Despite a significant reduction in Iba^+^ and Müller GFAP^+^ cells, PEDF treated mice still had a significantly higher number compared to controls. CTB was used as an anterograde tracer of RGC axons and this increase in CTB^+^ axons was only significant at 0.25 mm from the injury site and comparable to GFP only at further distances.
d. AAV2-CR2-Crry gene therapy Bosco A. et al., 2018 [24]	United States	DBA/2J mouse model of glaucoma	25 control, 26 GFP only, 34 treatment	IOP measurement RGC degeneration rates measured via Brn3a Axon fasciculations Optic nerve pathology via light microscopy images	Mice that were treated did not have any difference in expected IOP elevation compared to aged control and GFP only mice. Mice that were treated had significant differences in Brn3a^+^ densities compared to control and GFP only mice. Mice that were treated had significantly increased axon fasciculations compared to control and GFP only mice. Mice that were treated had significantly attenuated optic nerve pathology compared to control and GFP only mice.	DBA/2J mice develop progressive RGC degeneration as they age. These differences are based on Brn3a^+^ density and categorized into pre-degenerative, declining, and degenerative. Treated mice seemed to have an increased proportion and number of cells in the pre-degenerative and declining category, but these absolute numbers were not separately assessed. There was no significant difference between control and GFP only mice. Optic nerve pathology was assessed through light microscopy sections and may have subjective measuring as an established marker for ON pathology was not used.
e. AAV-STC-1 gene therapy Roddy GW. et al., 2017 [25]	United States	Transgenic mice models of retinal degeneration (P23H-1 and S334ter-4)	7 negative control P23H-1 mice, 8 negative control S334ter-4 mice, 7 treatment + P23H-1 mice, 8 treatment + S334ter-4	Optic nerve length (ONL) thickness and ERG responses Gene expression changes	Both P23H-1 and S334ter-4 treated mice had significantly thicker ONL compared to untreated transgenic mice. However, only P23H-1 treated mice had significant improvement in scotopic and photopic B-waves in ERG responses compared to untreated P23H-1 mice. S334ter-4 treated mice had significant gene expression changes with many genes involved in neuroprotection and anti-ROS.	Only P23H-1 treated mice showed improvement in ERG responses. These same genes were not upregulated in P23H-1 mice.
f. AAV2-Neuroglobin (NGB) gene therapy Cwerman-Thibault H. et al., 2017 [26]	France	DBA/2J mouse model of glaucoma	31 control (untreated), 18 early treatment (2 months), 12 late treatment (8 months)	RGC count via Brn3a^+^ cells Retinal stress measured via GFAP Morphological changes in neurons measured via Brn3a and β3-tubulin	Early treatment with AAV2-NGB significantly increased RGC counts compared to control. GFAP fluorescence intensities were reduced in both early and late treatments compared to untreated control. Early treatment appeared to maintain RGC density and dendrite branching, comparable to young untreated mice. In late treatment, dendritic profiles were better preserved than untreated and age matched mice.	Late treatment showed no increase in RGC counts compared to control. GFAP is used as a marker for retinal stress. Despite using actual markers to potentially assess fluorescence intensity or physical counts, these results were qualitative in nature.
g. AAV2-serum soluble Fas Ligand (sFasL) gene therapy Krishnan A. et al., 2016 [27]	United States	DBA/2J mouse model of glaucoma or microbead induction of elevated IOP in mice	10 negative control, 10 positive, control 10 GFP only, 10 treatment	IOP measurement RGC density and axon density Pro-apoptotic and anti-apoptotic mediators RGC density and axon density	There was no significant difference in IOP at months 3, 5, 7 or 9 (all measured time points) between positive control, GFP only, and treatment groups. RGC density and axon density were significantly increased with treatment compared to positive control and GFP only mice. Treated mice had significantly reduced mRNA levels of pro-apoptotic mediators, Fas, FADD, and BAX compared to GFP only. Treated mice had significantly increased mRNA levels of anti-apoptotic mediators cFLIP, Bcl2, and cIAP2 compared to GFP only. RGC density and axon density were significantly increased with treatment compared to positive control and GFP only mice.	Treatment had no improvement on IOP, and qualitatively, there was no improvement in pigment dispersion and iris atrophy. This increase in RGC and axon density was not significantly different from the negative control, suggesting protection comparable to normal levels. GFAP and TNFα levels were also reduced, and along with qualitative confocal microscopy of Müller glial cells, suggest reduced Müller glial cell activation in treated mice compared to GFP only. RGC and axon density was assessed in a microbead induced elevation of IOP model.

**Table 5 biomolecules-11-00581-t005:** Summary of in vivo studies utilizing microRNA as gene therapy.

Reference/Study	Country	Injury Model and Animal	Groups and Sample Size	Parameters for Efficacy	Main Result	Remarks
**5. microRNA gene based therapy**						
a. MicroRNA-21 based gene therapy **Li H.J. et al., 2018 [28]**	China	ON crush model, rats	6 negative control, 6 injury only, 6 injury + agomir (miR-21 mimic), 6 injury + antagomir (miR-21 inhibitor)	Axon count and flash visual evoked potentials (F-VEP) amplitude and latency RGCs count	Injury + antagomir rats had significantly higher axons/mm compared to negative control at all measured distances away from ON injury (250, 500, and 1000 µm). F-VEP amplitude was significantly higher in injury + antagomir group compared to injury + agomir groups. There was no significant difference between the two in F-VEP latency measurements. Injury + antagomir rats had significantly increased RGCs compared to injury only.	F-VEP amplitude is associated with functional optic nerve fibres and latency is associated with optic nerve demyelination or conductive issues. The significance between agomir and antagomir was not assessed or did not reach significance, despite both separately having significantly lower and higher RGC counts, respectively, compared to injury only rats.

## Data Availability

The authors agree to make all materials, data, and associated protocols promptly available to readers without undue qualifications in material transfer agreements.

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
