# Peer review of "Use of Gene Therapy in Retinal Ganglion Cell Neuroprotection: Current Concepts and Future Directions"

_biomolecules, 2021, doi:10.3390/biom11040581_

Round 1

Reviewer 1 Report

The manuscript is a systematic review concerning a well defined important subject. Source selection is properly explained.

The Authors shall mention that there are 2 types of glaucoma, open angle and closed angle. This is important because closed angle glaucoma is in most cases a medical emergency, likely not amenable to gene therapy.

The tables nicely summarize experimental results concerning preclinical research on gene therapy for retinal ganglion cell neuroprotection. In the narrative part of the paper presentation of rationale for each particular gene therapy approach would make the paper more interesting and readable. Just a few examples: To enhance neuroprotection in retina transfer of BDNF gene is being used, and few words about the role of BDNF and TRKbeta receptors in retina would help the reader to comprehend justification of this approach and its advantage compared to delivery of BDNF. Erythropoietin in the retina is mainly secreted by the Muller cells and functions in a paracrine manner to maintain cellular homeostasis. Is there any data concerning which cells in the retina express EPO following this gene therapy?  Stanniocalcin-1 is neuroprotective because of its involvement in regulation of cellular calcium/phosphate homeostasis. Etc…

Refs 2 and 3 do not contain journal names.

The headings sholuld precede the tables.

The Authors should be more precise in their wording, and double-check for misspelling. For example, they wrote: Injection of an adeno-associated virus (AAV)-based gene therapy which expressed NeuroD1 was able to convert reactive astrocytes… AAV-based gene was indeed injected, but these were cells (not gene therapy) that expressed NeuroD1. Also, ischemic stroke money cortices doesn’t make sense, should be “monkey cortices” perhaps?

Author Response

Thank you very much for taking your time to review out manuscript. We appreciate how much time you took to effectively look over our manuscript and provide us with valuable feedback. 

We have taken the time to address the concerns and revisions as requested:

1) Both open angle and closed angle glaucoma are introduced in the introduction

2) Some more context and rationale is provided for each gene therapy approach has been added

3) Ref 2 and 3 - now Ref 2 and 4 (respectively) have their journals included

4) Heading now precede the tables

5) Spelling and wording have been revised

Again, thank you so much for your feedback and review. We really appreciate your time and effort put into our manuscript.

Reviewer 2 Report

The authors conducted an in-depth review of the literature focusing on gene-based therapies for retinal ganglion cell protection.  Of the 82 articles resulting from their initial search, 18 were reviewed.  

Comments:  Overall, this is a very well written and well described review, which provides new information.  The topic is of relevance and interest to the readers of the journal.  Minor corrections are needed:

1) page 2: change "money" to "monkey".

2) page 3: correct the spelling of "focusing".

3) page 16: correct the sentence: "There was a significantly reduction.." to " There was a significant reduction..".

Author Response

Thank you very much for taking your time to review our manuscript. We really appreciate your kind words and feedback!

1). We have corrected "money" to "monkey"

2). we have corrected the spelling of "focussing" to "focusing"

3). we have changed "there was a significantly reduction..." to "there was a significant reduction..."

All minor revisions are requested by the reviewer have been corrected and changed.
Again thank you very much for your time and feedback.
